# Adapting Segment Anything Models to Medical Imaging via Fine-Tuning without Domain Pretraining

**Kevin Li [1], Pranav Rajpurkar PhD [2]**

[1]Stanford University
[2]Harvard Medical School
kevinli8@stanford.edu, pranav_rajpurkar@hms.harvard.edu

## Abstract

Medical image segmentation is an important task in the context of medical care, with applications in diagnostic and treatment processes. Segment Anything (SAM), a generalist foundation model trained on a corpus of 11 million natural images, demonstrates limited adaptability to the medical domain in a zero-shot prompting context, but shows promise under parameter-efficient fine-tuning. MedSAM is a foundation model which adapts SAM to the medical domain via training on a diverse medical corpus consisting of different modalities (one million images of modality CT, MRI, CXR, etc). In this work, we evaluate the advantage of MedSAM over SAM for medical task-specific adaptation achieved via parameter-efficient fine-tuning. Our results demonstrate that MedSAM does not yield a consistent advantage over SAM in this setting. We also introduce a novel parameter-efficient approach, LoRaMedNet, which combines elements of previous fine-tuning methods to achieve greater flexibility of adaptation for SAM, and find that LoRaMedNet-adapted SAM attains the best performance. The implication of this finding is that generalist models like SAM can achieve superior adaptation to specific medical tasks even when compared to models with medical pre-training.

## Introduction

Medical image segmentation, a vital aspect of patient diagnosis and treatment, has greatly benefited from advancements in deep learning. A significant development in this field is the emergence of generalist AI models like Meta AI's Segment Anything (SAM), designed to handle diverse segmentation tasks (Kirillov et al. 2023). Despite its versatility, SAM's training on general imagery poses limitations for medical applications, where specific imaging characteristics, like the amorphous structure of tissues, are present (Huang et al. 2024). In particular, zero-shot performance on medical benchmarks lags behind that of specialist deep learning models (Roy et al. 2023).

Addressing this gap, Ma et al. (2023) introduced MedSAM, trained exclusively on medical datasets to cater to the unique challenges of medical image segmentation. MedSAM showed improved zero-shot capabilities and in particular cases rivals that of specialist models. However, it remains an open question whether MedSAM's specialized training translates into superior performance compared to SAM, particularly when both models undergo advanced fine-tuning techniques.

**Contribution** Our hypothesis is that a foundation model does not need medical pre-training to successfully adapt to a downstream medical task using fine-tuning. This hypothesis is motivated by the reasoning that the medical pre-training dataset size is too small to provide a significant advantage under task-specific adaptation. We prove our hypothesis by introducing LoRaMedNet, a novel fine-tuning technique combining Low-Rank Adaptation (LoRA) of the SAM image encoder with a ConvNet prediction head. In particular, using the LoRaMedNet approach on SAM yields higher performance than that using LoRaMedNet on MedSAM, and also a higher performance than that of previous fine-tuning approaches on either SAM or MedSAM. This shows that for certain fine-tuning approaches, the advantages of medical pre-training are not clear. The important implication is that in the context of medical task-specific adaptation, generalist models like SAM need not be abandoned in favor of models with medical pre-training.

## Related Work

**Large Vision Model** After the release of SAM, there have been many papers on the adaptation of SAM to medical image tasks. The most direct method of adapting SAM has been by prompting it. Zero-shot prompting performance generally falls short of specialist models (Roy et al. 2023).

**Parameter-Efficient Fine-tuning** Fine-tuning approaches have seen greater success, with examples including ConvNet/ViT prediction head, LoRA adaptation of image encoder, and custom "$g$-network" prompt encoder (Hu, Xu, and Shi 2023; Zhang and Liu 2023; Shaharabany et al. 2023). These methods achieve state-of-the-art performance for foundation model based approaches, and in certain cases, as in the case of $g$-network, rival specialist models on benchmarks. The common approach is to keep the original SAM image encoder weights frozen, while fine-tuning the rest of the model which accounts for a small percentage of the total number of parameters. In particular it is possible to achieve state-of-the-art performance on downstream medical tasks using parameter-efficient fine-tuning

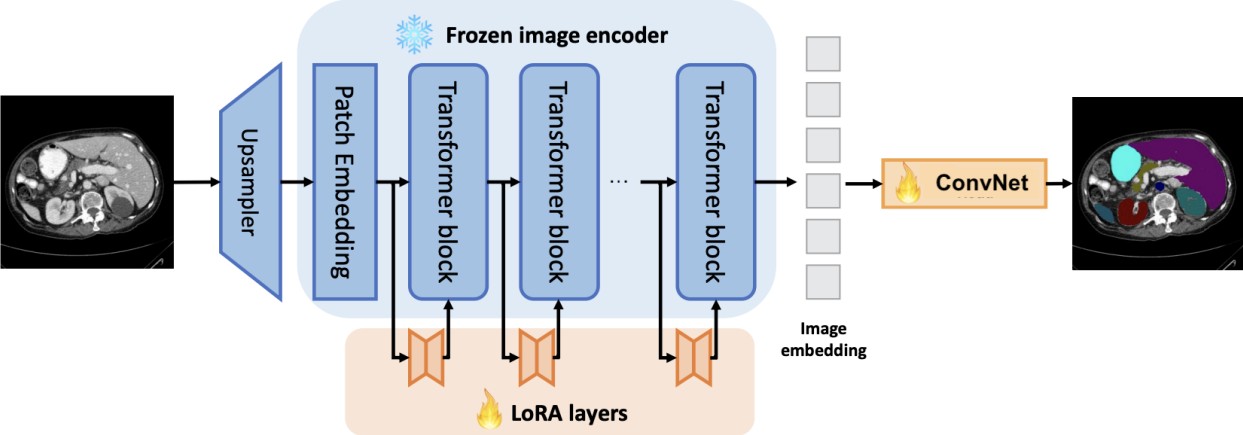

Figure 1: The LoRaMedNet Approach (diagram adapted from Zhang and Liu (2023))

of SAM, underscoring the potential of foundation models with natural image pre-training for domain adaptation to medical contexts.

**Medical Foundation Model**  MedSAM is a medical image segmentation foundation model adapting the SAM base (vit_b) model. SAM was adapted to the medical domain by fine-tuning the image encoder and mask decoder on a large corpus of over one million medical images consisting of various modalities, including CT scans, MRIs, CXR, Ultrasound, and Endoscopy (Ma et al. 2023). MedSAM's zero-shot performance on a general array of medical benchmarks exceeds previous segmentation foundation models and, in certain cases, exceeds specialist models.

The authors of the paper did not investigate the performance of MedSAM when fine-tuned for specific tasks, in particular whether MedSAM's medical pre-training was advantageous under parameter-efficient fine-tuning for task-specific adaptation. In particular, while the MedSAM training dataset is sizable and has a smaller domain gap with specific medical benchmarks we are interested in, it has only around one million image-mask pairs, which is dwarfed by the one billion masks in the SA-1B dataset used for SAM (Kirillov et al. 2023). Our hypothesis that MedSAM does not yield a consistent advantage over SAM relies on the idea that this medical pre-training adaptation is insignificant compared to the adaptation that occurs during task-specific fine-tuning, meaning that using MedSAM instead of SAM may not yield significantly better results. In our work we seek to test this hypothesis.

## Methods

**Foundation Model Comparison**  To understand the effect of using MedSAM instead of SAM under fine-tuning, we utilize previously considered parameter-efficient methods on MedSAM and compare to performance with SAM on a medical benchmark. This allows us to test if MedSAM yields a consistent advantage over SAM in the context of

task-specific adaptation. Since our focus is on task-specific adaptation, it makes sense to focus on automatic segmentation models which do not require any prompt. One motivation for having an automatic model and not a prompt-able one is that for promptable models, segmentation quality would then depend on prompt quality derived from domain expertise, which cannot be guaranteed in real-world scenarios. This is why the parameter-efficient methods we chose to evaluate on MedSAM all lead to automatic models.

To summarize, we evaluate performance using both SAM and MedSAM (vit_b model) image encoder and weights. The fine-tuning approaches we evaluate include the ConvNet and ViT prediction head from Hu, Xu, and Shi (2023) and the $g$-network method from Shaharabany et al. (2023).

**Novel Fine-tuning Method**  We also propose LoRaMedNet, which merges a LoRA adaptation of the SAM image encoder with a ConvNet prediction head, following the principles outlined in Hu, Xu, and Shi (2023). The LoRA adaptation of the image encoder follows Zhang and Liu (2023). The motivation for this approach is that we want a method to inject medical knowledge into the SAM image encoder, while also training a lightweight decoder network from scratch to automatically produce masks from the image embedding. While LoRaMedNet does train more parameters as it combines previous approaches, it is still parameter-efficient. The conceptual framework of LoRaMedNet is depicted in figure 1.

We designed this approach in an effort to maximize the flexibility of domain adaptation available to SAM in an effort to increase its performance, while still being parameter-efficient. The motivation for focusing on SAM is to understand the model's performance ceiling when compared to MedSAM. In particular, maximizing SAM's performance will present a challenge to the idea that MedSAM will always obtain superior performance, which is relevant to our hypothesis.

| Method | RV Dice | Myo Dice | LV Dice | Avg Dice |
|---|---|---|---|---|
| CNN Head (SAM) | 59.87 | 62.81 | 78.96 | 67.21 |
| CNN Head (MedSAM) | 65.45 | 77.13 | 90.06 | 77.55 |
| ViT Head (SAM) | 58.48 | 62.18 | 80.58 | 67.08 |
| ViT Head (MedSAM) | 59.80 | 61.12 | 79.24 | 66.72 |
| $g$ Network (SAM) | 57.92 | 66.23 | 84.04 | 69.40 |
| $g$ Network (MedSAM) | **78.58** | 82.02 | 91.71 | 84.10 |
| LoRaMedNet (ours) (SAM) | 77.80 | **84.06** | **94.15** | **85.34** |
| LoRaMedNet (ours) (MedSAM) | 73.68 | 80.90 | 94.01 | 82.86 |

Table 1: Comparison of fine-tuning methods on SAM and MedSAM Weights

**Existing Fine-Tuning Baselines**   Here we outline the details of previous fine-tuning methods, some of which we directly use on MedSAM. It is relevant for understanding LoRaMedNet, which combines two of these approaches.

Hu, Xu, and Shi (2023) fine-tune a novel prediction head, either a Convolutional Network or Vision Transformer that decodes the output of SAM's frozen image encoder. We elaborate more on the ConvNet head, which is used as part of LoRaMedNet. The ConvNet head follows the decoder structure of U-Net (Ronneberger, P.Fischer, and Brox 2015). The image embedding output first reshaped to $(256, 64, 64)$. This prediction head is then designed to upscale the feature map by 4x. LoRaMedNet utilizes a ConvNet head with depth 4 as described in the paper. The ConvNet prediction head serves as a benchmark in using advanced neural network architectures to decode the SAM image embedding, which we can compare LoRaMedNet to.

Zhang and Liu (2023) fine-tune SAM with Low-Rank Adaptation (LoRA) for the image encoder (Hu et al. 2021). LoRA is applied to query and value projection layers, and is applied to every transformer block in the SAM image encoder. See figure 2 for LoRA design. This serves as a benchmark for adapting SAM's image encoder for a downstream task by injecting medical knowledge.

Shaharabany et al. (2023) replace and fine-tune SAM's prompt encoder with a custom '$g$-network', tailored for image processing. This method achieves performance which surpasses previous state-of-the-art specialist models on various medical benchmarks, including nuclei, gland, and polyp segmentation. Hence $g$-network, which already achieves high performance with SAM, should present a challenge to MedSAM. Our goal with LoRaMedNet is to surpass $g$-network's performance in order to get closer to the performance ceiling with SAM.

**Dataset**   We employ the ACDC (Automated Cardiac Diagnosis Challenge) dataset, consisting of MRI scans of cardiac structures (Bernard et al. 2018), as the basis for our experiments. We chose this dataset for its applicability in testing segmentation models in the medical imaging arena.

The dataset consists of 100 patients, with two 3D volumes for each patient, and segmentation labels for three classes we are interested in—left ventricle, right ventricle, and myocardium. Following Hu, Xu, and Shi (2023), patients are split into train-validation-test with 70-15-15 ratio. For preprocessing, pixels within a volume are normalized to zero

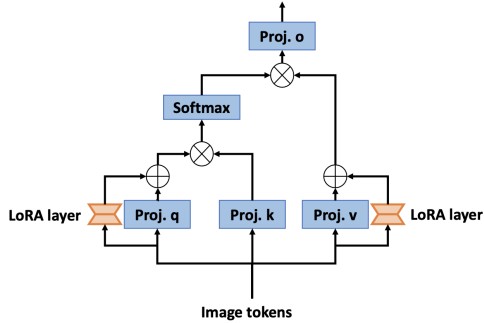

Figure 2: LoRA Adaptation (taken from Zhang and Liu (2023))

mean and unit variance, converted to RGB, and slices within a volume are used as 2D image input to the model. Our fine-tuning emphasizes label-efficiency, utilizing only 5 labeled volumes from a total of 70 training volumes. We assess performance based on the average Dice score across the three classes.

**Implementation**   We utilize the PyTorch codebase provided by Hu, Xu, and Shi (2023), making necessary modifications to implement our novel fine-tuning method, which in particular involves utilizing part of the codebase provided by Zhang and Liu (2023) for LoRA adaptation of image encoder.

Following Hu, Xu, and Shi (2023)'s setup for the ACDC dataset, we utilize Adam optimizer with learning rate $0.0005$, $(\beta_1, \beta_2) = (0.5, 0.999)$, batch size of $4$, and $120$ epochs after which convergence is achieved (Kingma and Ba 2017). During training, which was done on a single Tesla M40 with 24GB memory, data augmentation on images included Gaussian noise, brightness alteration, elastic deformation, and rotation. Loss function is given by the sum of cross entropy loss and dice loss (Sudre et al. 2017).

## Results

**Foundation Model Comparison**   In the comparison between MedSAM and SAM, we observed that MedSAM did not consistently outperform SAM. While it showed improvements in segmentation accuracy with the ConvNet head and $g$-network, there were no significant benefits with

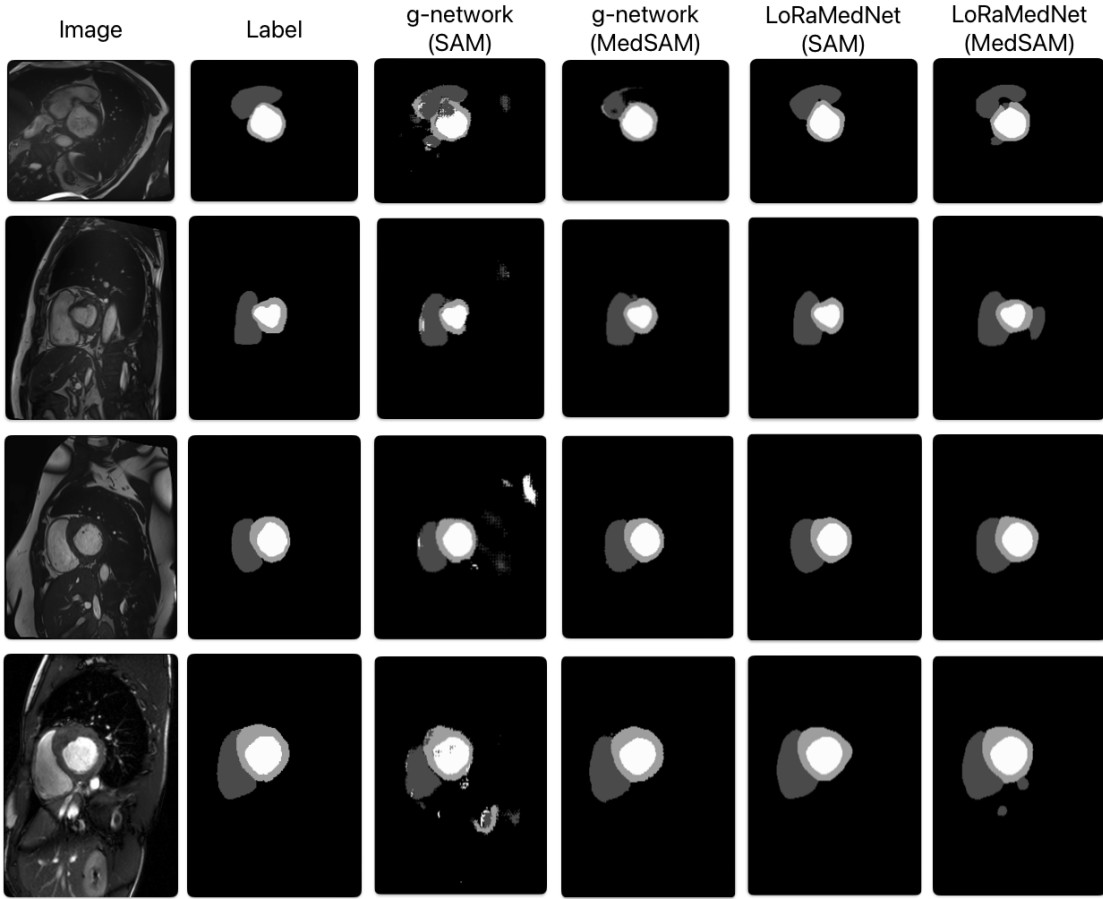

Figure 3: Comparison of Prediction Quality Across Methods

the Vision Transformer (ViT) head, and a slight decrease in performance was noted with LoRaMedNet.

**Fine-tuning Method Comparison**  LoRaMedNet, when fine-tuned with the SAM base model, demonstrated superior performance compared to previous fine-tuning methods, including those using MedSAM. With SAM, the next best approach was $g$-network, with remaining ConvNet and ViT head achieving comparable performance. With MedSAM, $g$-network manages to have a slight gain in performance over LoRaMedNet, with ConvNet now outperforming ViT head significantly. Full table of results is displayed in table 1.

**Prediction Quality**  Figure 3 shows a side-by-side comparison of predictions based on different fine-tuning methods on different foundation models. We observe that LoRaMedNet-adapted SAM has consistently superior performance which corroborates our table results. In particular LoRaMedNet outperforms $g$-network, a strong baseline method which suffers from grainy predictions. This is the most apparent with $g$-network-adapted SAM, and the effect of using MedSAM instead seems to be a reduction in this graininess. On the other hand, considering LoRaMedNet-adapted SAM which already has high performance, using MedSAM instead does not yield any noticeable advantage

and in fact seems to degrade performance.

**Discussion**  Our hypothesis is supported by the observation that MedSAM does not yield consistent advantages over SAM in performance, and that the highest performing method, LoRaMedNet (ours), utilized SAM. The success of LoRaMedNet as a parameter-efficient fine-tuning approach is significant, given the high benchmarks set by recent methods like the $g$-network. Furthermore, LoRaMedNet achieves consistent high performance on both SAM and MedSAM, whereas previously considered methods achieve lower performance with at least one of these two model weights. This demonstrates the robustness of using LoRaMedNet on foundation models with or without medical pre-training. Our results also demonstrate the feasibility of attaining high performance in a label-efficient setting.

These results carry important implications for medical imaging. They indicate that generalist AI models, such as SAM, can be effectively and efficiently (both parameter-wise and label-wise) adapted for medical image segmentation tasks, even without specific pre-training on medical datasets (Moor et al. 2023). The effectiveness of LoRaMed-Net with SAM underscores the potential for more adaptable and cost-efficient AI solutions in healthcare, which could lead to enhanced patient diagnosis and treatment.

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
