# OpenReview forum: "Adapting Segment Anything Models to Medical Imaging via Fine-Tuning without Domain Pretraining"
_AAAI.org/2024/Spring_Symposium_Series/Clinical_FMs — AAAI 2024 SSS on Clinical FMs_

### Official Review · Reviewer_mT6h · 2024-02-16

**Rating:** 7
**Confidence:** 4

**Review:**

This paper proposes a variant of low rank adaptation method for SAM and MedSAM models to medical imaging tasks.

Strengths
* The effectiveness of the adaptation on SAM is noteworthy, demonstrating significant improvements even without extensive pretraining or finetuning in the medical domain
* The clarity and comprehensiveness of the writing make the methodology, experiments, and results easily understandable
* The comparative analysis is included, benchmarking against current baselines and showcasing the proposed method's advantages clearly.

Weaknesses
* Considering LoRA is already a popular approach as a parameter efficient fine-tuning method, the novelty of method is limited.

---

### Official Review · Reviewer_3GuJ · 2024-02-21
**Parameter efficient fine-tuning SAM for medical image segmentation**

**Rating:** 6
**Confidence:** 4

**Review:**

Summary of Contributions:

The work proposes a new way to efficiently fine-tune SAM for medical image segmentation, i.e. a custom lightweight ConvNet head after the SAM encoder. The SAM encoder undergoes parameter efficient fine-tuning by using Low Rank Adaptation. The authors claim and demonstrate that fine-tuning SAM is better than fine-tuning MedSAM, i.e. for successful adaptation, medical pretraining is not necessary.

Strengths:
1. The authors challenge the popular belief that foundation models should be pre-trained with data from the domain where they are to fine-tuned, and successfully demonstrate that (in their own words) “generalist models like SAM need not be abandoned in favor of models with medical pre-training”
2. The work experiments with various decoder networks for fine-tuning SAM and MedSAM.
3. The paper is easy to follow and the Block Diagrams are representative of the methods provided in the work.
4. Preprocessing steps, training hyper-paramters, along with memory requirements have been provided.

Weaknesses:
1. The work does not provide quantitative comparisons with specialist models (i.e. models trained from scratch on the given dataset, say UNet, which inspired the Convent decoder)
2. The work only provides fine-tuning results for a single dataset. Ideally two more medical image segmentation dataset (such as BraTS, VerSE, etc.) should be included.

Questions, Suggestions, Comments:
1. What are the input and output dimensions for the images?
2. It is not very clear that how the decoder from UNet is used, more specifically: the UNet decoder uses a prior from the encoder after each upscaling step (in the form of residual connections). What priors (if any) are being used in the proposed decoder ConvNet?
3. Addressing the weakness will definitely improve the quality of the work, and make the author’s claim more substantiated.

---

### Official Review · Reviewer_dpBW · 2024-02-22

**Rating:** 9
**Confidence:** 4

**Review:**

The paper addresses the challenge of adapting generalist foundation models like SAM to medical image segmentation tasks, crucial for diagnostic and treatment processes. Despite initial promise, the specific adaptation model, MedSAM, fails to consistently outperform SAM in medical task-specific contexts. However, the introduction of LoRaMedNet, a novel parameter-efficient approach, demonstrates the potential for enhancing SAM's adaptability and achieving superior performance in medical tasks. The findings underscore the significance of exploring adaptation techniques for generalist models, suggesting that they can excel in specific medical applications even without dedicated medical pre-training.

---

### Official Review · Reviewer_rHvS · 2024-02-22
**Relevant contribution showcasing the advantage of generalist foundation models**

**Rating:** 8
**Confidence:** 2

**Review:**

This paper evaluates the usage of in-domain pretraining for medical segmentation tasks as opposed to the use of a generalist foundation model for segmentation (SAM), and proposes a new PEFT technique: LoRaMedNet. The authors show that the in-domain pretrained model, MedSAM, does not consistently outperform SAM when fine-tuning on medical segmentation data, which brings the usefulness of MedSAM into question. Additionally, their proposed LoRaMedNet outperforms standard PEFT techniques for fine-tuning on medical data. The technique combines standard LoRA with an additional convolutional head. The set of comparisons also seem solid as a workshop contribution, as they compare to a variety of existing fine-tuning baselines on their target dataset -- the Automated Cardiac Diagnosis Challenge (ACDC). This paper seems quite relevant for the workshop and is well-written.